# Association between Hyperglycemia and Medication-Related Osteonecrosis of the Jaw (MRONJ)

**DOI:** 10.3390/jcm12082976

**Published:** 2023-04-19

**Authors:** Gabor Kammerhofer, Daniel Vegh, Dorottya Bányai, Ádám Végh, Arpad Joob-Fancsaly, Peter Hermann, Zoltan Geczi, Tamas Hegedus, Kata Sara Somogyi, Bulcsú Bencze, Zita Biczó, Donát Huba Juhász, Péter Zaborszky, Márta Ujpál, Mihály Tamás Vaszilkó, Zsolt Németh

**Affiliations:** 1Department of Oromaxillofacial Surgery and Stomatology, Semmelweis University, 1088 Budapest, Hungary; 2Diabetes-Dental Working Group, Semmelweis University, 1088 Budapest, Hungary; vegh.daniel@semmelweis.hu (D.V.); bencze.bulcsu@dent.semmelweis-univ.hu (B.B.);; 3Department of Prosthodontics, Semmelweis University, 1088 Budapest, Hungary; 4Department of Paediatric Dentistry and Orthodontics, Semmelweis University, 1088 Budapest, Hungary; 5Department of Oral Diagnostics, Semmelweis University, 1088 Budapest, Hungary; 6Faculty of Dentistry, Semmelweis University, 1088 Budapest, Hungary

**Keywords:** diabetes, hyperglycemia, osteonecrosis, bisphosphonates, oral surgery

## Abstract

Background: Medication-related osteonecrosis of the jaw (MRONJ) is a type of jawbone necrosis caused by the use of drugs for some types of cancer and osteoporosis. The current study aimed to evaluate the associations between hyperglycemia and the development of medication-related osteonecrosis of the jaw. Methods: Our research group investigated data collected between 1 January 2019 and 31 December 2020. A total of 260 patients were selected from the Inpatient Care Unit, Department of Oromaxillofacial Surgery and Stomatology, Semmelweis University. Fasting glucose data were used and included in the study. Results: Approximately 40% of the necrosis group and 21% of the control group presented with hyperglycemia. There was a significant association between hyperglycemia and MRONJ (*p* < 0.05, *p* = 0.003). Vascular anomaly and immune dysfunction caused by hyperglycemia can lead to necrosis after tooth extraction. Necrosis is more common in the mandible (75.0%) and in the case of parenteral antiresorptive treatment (intravenous Zoledronate and subcutaneous Denosumab). Hyperglycemia is a more relevant risk factor than bad oral habits (26.7%). Conclusions: Ischemia is a complication of abnormal glucose levels, a possible risk factor for necrosis development. Hence, uncontrolled or poorly regulated plasma glucose levels can significantly increase the risk of jawbone necrosis after invasive dental or oral surgical interventions.

## 1. Introduction

Medication-related osteonecrosis of the jaw (MRONJ) is a chronic and multifactorial inflammatory disease that occurs in patients with a history of either previous or current antiresorptive therapy [1]. The first documented case was reported in 2003 by Marx and his colleagues, which was attributed to Pamidronate and Zoledronate [2]. In the case of MRONJ, one or more necrotic bone surfaces may be visible in the maxillofacial region, either through visible inspection or intra- or extraoral fistulas, and persist for a minimum of eight weeks without responding to appropriate therapy [3]. Furthermore, it is important to rule out that the abnormality has arisen due to radiation or as a metastasis of a malignant process [1].

The two main elements of antiresorptive therapy are bisphosphonates and Denosumab injection [3]. Bisphosphonates are pyrophosphate derivatives with a half-life of several years that strongly bind to hydroxyapatite in the bone [4]. Currently used aminobisphosphonates can be released from bones degraded by osteoclasts and trigger cellular apoptosis by inhibiting the farnesyl pyrophosphate synthase enzyme [5]. Denosumab, on the other hand, is a human monoclonal antibody that targets the receptor activator of nuclear factor-κB ligand (RANKL), which plays a significant role in the formation and survival of osteoclasts [5,6]. Bisphosphonates and denosumab are primarily used to treat malignant oncological metastatic diseases and osteoporosis [5,7].

We evaluated the classification of MRONJ stages based on the position paper issued by the American Association of Oral and Maxillofacial Surgeons (AAOMS) in 2022. According to this classification, there are five stages, each briefly described. The first one is “patients at risk”. This stage includes patients with a history of antiresorptive therapy but no symptoms [8]. Stage 0 is characterized by the absence of visible necrotic bone surface, but the presence of some nonspecific symptoms (such as odontalgia, aching bone, and numbness) and/or clinical (spontaneous tooth loss and mucosal swelling) or radiological (alveolar bone loss, sclerotic bone transformation, and thickening of the periodontal space) findings [8]. In stage one, dead bone or fistula in the oral cavity is already visible, but there is no sign of inflammation. In stage 2, the evidence of infection/inflammation can be observed next to the necrotic bone surface or fistula [8]. Moving on to stage 3, in addition to the findings mentioned above, at least one of the following lesions will be present: pathological bone fracture, extraoral fistula, oro-nasal/oro-antral communication, bone necrosis extending to the mandibular base, maxillary sinus base, or zygomatic bone. MRONJ can be confused with various other conditions (such as alveolar osteitis, sinusitis, atypical neuralgia, chronic sclerosing osteomyelitis, fibro-osseous lesions, and sarcoma), so a thorough history-taking, clinical examination, and other imaging procedures are important for an accurate diagnosis [8].

Medication-related osteonecrosis of the jaw can be caused by various risk factors, including the type of antiresorptive drugs used and the mode and duration of their administration. The probability of developing MRONJ is increased with intravenous (IV) medication compared to oral dosing. The followings can act as a trigger in developing MRONJ. In addition, the presence of inflammatory diseases in the oral cavity, such as periodontitis, peri-implantitis, or periapical inflammation, as well as dental procedures such as extractions, implantation, and other periodontal interventions performed as a result of inflammation, are also considered risk factors [8]. Increased occurrence has been detected in patients with removable dentures, smokers, and also in the lower jaw. Women are also more likely to develop MRONJ attributed to gender-specific underlying diseases such as breast cancer and postmenopausal osteoporosis. Other risk factors include malignant diseases, systemic diseases, and medications such as antiangiogenic drugs [8,9,10]. They have already described cases where unsuccessful or faulty endodontic treatment (over- or under-filled root, root fracture, or root perforation) was the only reported triggering factor in the development of MRONJ [11].

Hyperglycemia is a crucial healthcare and social issue that we face today. It has become a widespread concern in developed industrial societies and can be classified as a folk disease. As such, it is also recognized as a folk disease in Hungary, as evidenced by studies [12,13]. Given the gravity of this issue, it is imperative that dentists, as members of the medical community, familiarize themselves with the dental characteristics of this civilization’s disease. Specifically, the periodontium, alveolar process, and microcirculation are the most vulnerable areas, as they are at risk of extensive damage due to hyperglycemia [14,15]. When it comes to MRONJ and diabetes mellitus (DM), wound healing becomes more challenging and damaged. Despite some literature investigating the relationship between hyperglycemia and MRONJ, there still needs to be a consensus. Therefore, this publication aims to enhance our understanding of this association. Although the exact mechanism by which DM may promote MRONJ has yet to be determined, it is well-known that DM is a risk factor for this complication [16,17,18]. Ischemia, a complication of abnormal glucose levels, is also a potential risk factor for MRONJ development in DM patients. It is essential to note that uncontrolled or poorly regulated plasma glucose levels can significantly increase the risk of jawbone necrosis following invasive oral surgical interventions.

One of the primary aims of our research is to broaden our understanding of knowledge that can potentially aid in the prevention of MRONJ and enhance the quality of life for patients. Previous studies have shown that hyperglycemia plays a significant role in developing and treating oral lesions. Additionally, our previous examinations have revealed an association between oral tumors and hyperglycemia [19]. Thus, further research on MRONJ should be conducted based on the relationship between high plasma glucose levels and malignant lesions.

The potential association between hyperglycemia and MRONJ has important clinical implications. Healthcare providers should be aware of this potential association when treating patients with diabetes or other conditions associated with hyperglycemia who are at risk for developing MRONJ. These patients should be carefully monitored for the development of MRONJ, and preventative measures, such as proper dental care and avoiding unnecessary dental procedures, should be taken.

Additionally, healthcare providers should consider screening patients with MRONJ for hyperglycemia, as this condition may be an indicator of poor bone healing and increased susceptibility to infections. Patients with MRONJ and hyperglycemia may benefit from more aggressive treatment approaches, such as the use of antibiotics or surgical intervention, to improve bone healing and prevent the development of infections

The clinical therapy of MRONJ can be challenging due to the lack of a standardized treatment protocol. However, several approaches have been proposed for the management of this condition, which primarily focus on controlling pain, infection, and promoting bone healing.

**Conservative therapy**:

Conservative therapy is the initial approach to manage MRONJ in most cases. This includes the following:*Antibiotics:* Antibiotics are usually prescribed to manage bacterial infections associated with MRONJ. Antibiotics such as penicillin, clindamycin, and metronidazole are commonly used. The duration of antibiotic therapy is typically 7–14 days.*Pain management:* Pain management is an important aspect of MRONJ therapy. Nonsteroidal anti-inflammatory drugs (NSAIDs) and opioids are commonly used to manage pain associated with MRONJ. However, opioids are generally avoided due to the risk of addiction.*Chlorhexidine mouthwash:* Chlorhexidine mouthwash is used to manage oral infections and prevent the progression of MRONJ.*Local debridement:* Local debridement is the removal of necrotic bone tissue in the affected area. This procedure is performed under local anesthesia and may be repeated several times until the area is free of necrotic tissue [8].


**Surgical therapy:**


Surgical therapy may be necessary for patients with advanced MRONJ or those who have failed conservative therapy. The following surgical procedures are commonly performed for MRONJ: *Sequestrectomy:* Sequestrectomy is the removal of necrotic bone tissue to promote healing. This procedure is performed under general anesthesia and may require a hospital stay.*Segmental resection:* Segmental resection involves the removal of a segment of the jawbone affected by MRONJ. This procedure is reserved for severe cases and may require reconstruction of the jawbone using bone grafts. *Hyperbaric oxygen therapy:* Hyperbaric oxygen therapy involves breathing pure oxygen in a pressurized chamber to promote tissue healing. This therapy has been used in combination with surgical therapy to improve healing outcomes [8].

**Prevention**:

Prevention is an important aspect of MRONJ therapy. The following preventive measures are recommended for patients at risk of developing MRONJ:*Proper dental care:* Patients taking antiresorptive medications should receive regular dental checkups and maintain good oral hygiene to prevent infections.*Avoiding dental extractions:* Dental extractions should be avoided whenever possible in patients taking antiresorptive medications.*Drug holidays:* Drug holidays, or discontinuation of antiresorptive medications, may be recommended in some cases to reduce the risk of MRONJ. However, this approach is controversial and should be carefully considered on a case-by-case basis.

In conclusion, the clinical therapy of MRONJ can be challenging, and a multidisciplinary approach involving dentists, oral surgeons, and endocrinologists may be necessary. The choice of therapy depends on the severity of the condition and the individual patient’s response to treatment. Conservative therapy is usually the first approach, followed by surgical therapy for advanced cases. Prevention is an important aspect of MRONJ therapy and should be emphasized in patients at risk of developing this condition [8].

The current study aimed to evaluate the associations between hyperglycemia and the development of medication-related osteonecrosis of the jaw with retrospective analysis of our patient data.

## 2. Materials and Methods

We evaluated data collected between 1 January 2019, and 31 December 2020. In total, 260 patients admitted to the Department of Oromaxillofacial Surgery and Stomatology, Semmelweis University, were included in the study. Then, they were classified under the necrosis group, which comprised patients with MRONJ, and the control group, which included patients who required other medical care; planned maxillofacial trauma surgery, wisdom tooth extraction, and benign tumor or cyst surgery. Within the indicated period, we treated 200 control patients and 60 patients with MRONJ in our clinic. We registered every patient in our study whom we treated in the University Department. The position papers were developed by the American Association of Oral and Maxillofacial Surgeons (AAOMS, 2014; 2020, 2022) [8,20,21], appointed by the board and composed of clinicians with extensive experience in caring for these patients and basic science researchers. All patients with MRONJ who required surgical therapy in addition to conservative therapy were included in the study. Due to differential diagnosis problems and ethical considerations, some patients were excluded from the study. This provided more relevant information on glucose status and the adverse effects of hyperglycemia. We excluded from the study patients under 18 years of age who were unable to cooperate, pregnant women, people with chronic immunodeficiency, patients receiving systemic steroid therapy, patients receiving head and neck irradiation, and drug addicts.

In clinical practice, measuring fasting glucose levels from finger blood is still the most commonly used. Glucose levels were measured on admission to the clinic. Before blood sampling, patients fasted for 12 h, so the samples were taken on an empty stomach (fasting glucose). This was facilitated by the fact that blood was taken in the morning and by qualified medical staff. Normoglycemia was a plasma glucose level of <6.1 mmol/L. (Definition and diagnosis of diabetes mellitus and intermediate hyperglycemia: a WHO/IDF consultation report. World Health Organization 2006). 

Data analysis was performed using Prism version 8.4.2. (464, GraphPad Software, San Diego, CA, the USA). The Pearson’s chi-square test was used for statistical analysis. A *p*-value of <0.05 was considered significant. All data were stored using Microsoft Excel. 

Our study followed the ethical standards laid down in the 1964 Declaration of Helsinki and its later amendments and was approved by the Institutional Ethics Committee of Semmelweis University of Budapest (Semmelweis University Regional and Institutional Committee of Science and Research Ethics, protocol code: SE-RKEB 204/2018). 

Only members of the research team had access to the human data. The data were stored on a password-protected medium. The authors confirm that all methods were performed in accordance with the relevant guidelines and regulations. 

## 3. Results

The MRONJ (necrosis) group included 60 patients (39 women [65%] and 21 men [35%]). The average age of the patients was 65 years, and the standard deviation (SD) was 10.66. Approximately 40% of the examined patients presented with hyperglycemia. Meanwhile, the control group included 200 patients (72 women [36%] and 128 men [64%]). The average age of the patients was 50 years and SD was 19.29. Approximately 21% of the control group presented with hyperglycemia. There was a significant association between hyperglycemia and MRONJ (*p* < 0.05, *p* = 0.003). The necrosis group had a higher proportion of patients with hyperglycemia than the control group. MRONJ can more likely develop based on metabolic differences, which are caused by hyperglycemia, after invasive interventions involving the jawbone, surgeries, and tooth extractions. Table 1 depicts the blood glucose levels and sex of the participants.

Regardless of glucose balance, 75% (n = 45) of patients in the necrosis group developed necrosis in the lower jawbone. Painful bone necrosis developed only on the upper jawbone in 18.4% of patients (n = 11), and on the upper and lower jawbones in 6.6% (n = 4). However, the location difference was not significant in terms of glycemic levels (*p* > 0.05, *p* = 0.825). Next, we depicted the location of necrosis and the patient’s blood glucose level. MRONJ occurred more frequently on the lower jawbone (Table 2).

Approximately 38.3% (n = 23) of patients who were examined received anti-resorptive medical treatment due to breast carcinoma (CC.), 23.3% (n = 14) received anti-resorptive medical treatment due to prostate cc., 15% (n = 9) received anti-resorptive medical treatment due to osteoporosis, 10% (n = 6) received anti-resorptive medical treatment due to multiple myeloma, 8.3% (n = 5) received anti-resorptive medical treatment due to lung cc. and 5% (n = 3) received anti-resorptive medical treatment due to other malignant diseases (non-Hodgkin’s lymphoma, urothelial cc., and colorectal cc.). The number of patients who received bisphosphonate therapy due to breast and prostate cc. was the highest.

Antiresorptive medications such as parenteral bisphosphonates and Denosumab are the most common risk factors for developing MRONJ. Approximately 39.9% (n = 24) of participants in this study received this treatment. Furthermore, 21.7% (n = 13) of patients who were examined received subcutaneous (SC.) Denosumab. Approximately 11.8% (n = 7) of patients received both iv. Zoledronate and sc. Denosumab and 8.3% (n = 5) of the MRONJ group was treated with per os (P.O.) Ibandronate; 8.3% (n = 5), combined iv. Zoledronate, sc. Denosumab, and p.o. Ibandronate; 5% (n = 3), combined iv. Zoledronate, sc. Denosumab, and p.o. Clodronate; 3.3% (n = 2), p.o. Alendronate; and 1.7% (n = 1), p.o. Risedronate. We evaluated data on the primary disease and its corresponding medication, glucose balance, and necrosis location. We expressed the type of medicine used for each disease as a percentage. All patients with multiple myeloma and lung tumor received iv. Zoledronate therapy. Patients with osteoporosis were commonly treated with p.o. dosage. Overall, parenteral administration was dominant among the data (Table 3).

Based on recall data, hyperglycemia is among the three most significant risk factors, followed by smoking and alcohol consumption. Hyperglycemia was among the most relevant risk factor in 26.7% (n = 16) of patients with MRONJ (Table 4).

## 4. Discussion

Regarding the jawbone anatomy, the mandible has an end arterial blood supply. Thus, the collateral circulation is minimal [22], which indicates that tissue death caused by vascular anomaly develops more easily. Numerous studies have shown that several patients develop MRONJ after dental care because the lumen of the small vessels of the jaw is obliterated, thereby impeding the end arterial blood supply of the mandible. Consequently, ischemia and necrosis may occur. Due to the increased involvement of the lower jawbone and the lower stability of the lower plate prosthesis, a more frequent dental check-up of the lower prosthesis is recommended for patients. Regularly relining of the lower prosthesis is also recommended [23,24,25,26,27,28,29].

Disease control and patient education are essential due to prevention. We can prevent trauma to the mucosa bone foundation, the development of decubitus, and painful tissue injury by preventing these problems. Over one-third of patients who were examined presented with breast tumors, and almost one-third had prostate tumors. One-quarter of patients were diagnosed with osteoporosis or multiple myeloma. We did not identify patients with renal cc. who are receiving anti-resorptive treatment in our examination. Hence, it may arise as an opportunity for further research. One of the possible reasons for this is that patients treated with renal cell cc. commonly receive biological, targeted antiangiogenic therapy in recent years via the dynamic development of molecular oncology [30,31,32]. Hence, referring patients to a dentist or a mouth surgeon is important before starting oncological, anti-resorptive therapy.

Therefore, for sanitation, the dentist must take a general medical history. In addition, the attending physician should emphasize that the potential inconvenience of oral surgery rehabilitation and the degree of pain occurring at this time are lesser than the incidence of bone necrosis. Notably, mandibular involvement is a factor for both dysglycemia and normal blood glucose levels. Like normal plasma glucose levels, breast and prostate tumors should be an underlying disease in dysglycemia. Several female patients in the necrosis group presented with osteoporosis and breast tumor, commonly treated with bisphosphonate or other anti-resorptive drugs. The incidence of these diseases is more frequent in women than in men. In our study, all patients who presented with multiple myeloma were women. Findings on whether the use of iv. Zoledronate or sc. Denosumab is associated with a greater risk of MRONJ but remains inconsistent. However, most authors agree that using these active substances increase the risk of jawbone necrosis [33,34,35]. Our domestic research had a similar result. Among the factors mentioned above, abnormal glucose levels are the most significant risk factor for the development of MRONJ. Moreover, smoking and alcohol consumption can be risk factors for necrosis [16,17,36,37,38].

Vascular anomaly and immune dysfunction due to hyperglycemia can be a predisposing factor for the development of necrosis after tooth extraction. The risk of developing jawbone necrosis can be reduced by professional premedication, careful planning and execution of minimally invasive tooth removal and oral surgery, and appropriate postoperative care [36,37,38,39,40,41,42,43,44].

This review has some limitations. Our study design did not investigate oral health statuses, such as periodontal disease, dental prostheses, dental implants, and periodontal surgeries. In our study, we examined 60 patients with MRONJ and 200 controls. The difference is large, the reason being that we wanted to include as many patients as possible. Despite these limitations, this real-world observational study demonstrated that the risk of developing MRONJ was significantly higher in patients with dysglycemia. After examining 60 patients, it cannot be stated absolutely that hyperglycemia is one of the most relevant causes of MRONJ. However, it is an important predisposing factor.

The number of patients with both dysglycemia and anti-resorptive therapy is increasing. Therefore, patients with these conditions need medical care more frequently in dental offices. However, MRONJ can be treated, and the likelihood of developing this condition can be reduced through preventative dental care and the maintenance of good oral hygiene. As part of a multi-professional team, dentists have a critical role in preventing MRONJ. Oral health practitioners need to work closely with the patient’s medical oncology team to advocate for premedication evaluation to prevent MRONJ and identify potential risks of MRONJ and if the disease does develop despite the careful planning, it should be diagnosed as early as possible in order to avoid more significant damage. Taken together, these patients require close monitoring in a clinical setting.

Our other purpose is to continue and improve our work. In addition to the work of healthcare professionals, social cohesion is also necessary for successful treatment and care. Our team has set the fundamental purpose of the most comprehensive possible mapping of issues that arise during the dental care of patients with DM. Furthermore, our future studies must examine MRONJ relapse in the future. We believe that the study topic and knowledge of dental issues are increasingly important due to the increasing incidence of DM and MRONJ. Hence, further studies should be conducted. Further studies are planned using hemoglobin A1C (HbA1C.) measurements, although we have already measured HbA1C, the results of which have been published by our research team, so we have shown a method to follow [45].

## 5. Conclusions

In conclusion, the results of this study suggest that in patients with dysglycemia, the risk of developing MRONJ is statistically significantly higher than in patients with normoglycemia. This study evaluates the properties (e.g., age, sex, type of cancer, presence of osteoporosis) and habits (like smoking and drinking alcohol) of patients and the possible combined effects of dysglycemia and anti-resorptive therapy that could lead to ischemia. Based on the abovementioned findings, ischemia can be a potential risk factor for developing jawbone necrosis after invasive dental or oral surgical interventions, especially in the lower jaw. 

## Figures and Tables

**Table 1 jcm-12-02976-t001:** Prevalence of hyperglycemia and normoglycemia both in the case and in the control groups.

Sex	Glucose Status	Case Group		Control Group	
Male	Hyp	10	16.7%	27	13.5%
Male	Norm	11	18.3%	101	50.5%
Female	Hyp	14	23.3%	15	7.5%
Female	Norm	25	41.7%	57	28.5%
		60	100.0%	200	100.0%

Hyp: Hyperglycemia, Norm: Normoglycemia.

**Table 2 jcm-12-02976-t002:** The location of MRONJ and the glucose levels were expressed as percentages in the necrosis group. The association between dysglycemia and the location of MRONJ is shown.

Location	Glucose Status	Patients	
Maxilla	Hyp	4	6.7%
Maxilla	Norm	7	11.7%
Mandible	Hyp	18	30.0%
Mandible	Norm	27	45.0%
Both	Hyp	2	3.3%
Both	Norm	2	3.3%
		60	100.0%

Hyp: Hyperglycemia, Norm: Normoglycemia.

**Table 3 jcm-12-02976-t003:** Different parameters (primary disease and its associated medications, glucose balance, and location of necrosis in patients with MRONJ).

Primary Disease	Agent	Hyp	Norm	Mandible	Maxilla	Both	Patients	Total%	Illness%
Others	Zoledronate iv.	1	1	2	0	0	2	3.3%	66.7%
Others	Zoledronate iv. + Denosumab inj.	1	0	1	0	0	1	1.7%	33.3%
Breast cc. wbm.	Denosumab inj.	3	1	4	0	0	4	6.7%	17.4%
Breast cc. wbm.	Ibandronate p.o.	2	1	2	0	1	3	5.0%	13.0%
Breast cc. wbm.	Zoledronate iv.	3	5	7	1	0	8	13.3%	34.8%
Breast cc. wbm.	Zoledronate iv. + Denosumab inj.	1	0	1	0	0	1	1.7%	4.3%
Breast cc. wbm.	Zoledronate iv. + Denosumab inj. + Clodronate p.o.	0	2	2	0	0	2	3.3%	8.7%
Breast cc. wbm.	Zoledronate iv. + Denosumab inj. + Ibandronate p.o.	1	4	4	1	0	5	8.3%	21.7%
Multiple myeloma	Zoledronate iv.	3	3	4	1	1	6	10.0%	100.0%
Osteoporosis	Alendronate p.o.	1	1	2	0	0	2	3.3%	22.2%
Osteoporosis	Denosumab inj.	0	2	1	1	0	2	3.3%	22.2%
Osteoporosis	Ibandronate p.o.	0	2	2	0	0	2	3.3%	22.2%
Osteoporosis	Risendronate p.o.	0	1	0	1	0	1	1.7%	11.1%
Osteoporosis	Zoledronate iv. + Denosumab inj.	0	1	0	1	0	1	1.7%	11.1%
Osteoporosis	Zoledronate iv. + Denosumab inj. + Clodronate p.o.	0	1	1	0	0	1	1.7%	11.1%
Prostate cc. wbm.	Denosumab inj.	3	4	4	1	2	7	11.7%	50.0%
Prostate cc. wbm.	Zoledronate iv.	1	2	2	1	0	3	5.0%	21.4%
Prostate cc. wbm.	Zoledronate iv. + Denosumab inj.	3	1	3	1	0	4	6.7%	28.6%
Lung cc. wbm.	Zoledronate iv.	1	4	3	2	0	5	8.3%	100.0%
		24	36	45	11	4	60	100.0%	

Hyp: hyperglycemia, Norm: normoglycemia, cc. wbm.: carcinoma with bone metastasis.

**Table 4 jcm-12-02976-t004:** Characteristics (bad oral habits and glucose status) of the necrosis group.

Alcohol	Smoking	Glucose Status	Patients	
No	No	Norm	26	43.3%
No	No	Hyp	16	26.7%
No	Yes	Norm	5	8.3%
No	Yes	Hyp	4	6.7%
Yes	No	Norm	2	3.3%
Yes	No	Hyp	3	5.0%
Yes	Yes	Norm	3	5.0%
Yes	Yes	Hyp	1	1.7%
			60	100.0%

Hyp: Hyperglycemia, Norm: Normoglycemia.

## Data Availability

The datasets used and analyzed during the current study available from the corresponding author on reasonable request.

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
