# Peer review of "Association between Hyperglycemia and Medication-Related Osteonecrosis of the Jaw (MRONJ)"

_jcm, 2023, doi:10.3390/jcm12082976_

Round 1

Reviewer 1 Report

The article presents the study related to diysglicemia and medication-related osteonecrosis of the jaw. Alsthough, in the pandemic of diabetes and its negative consequances on the health quality, also oral, the article needs some corrections and cannot be published in the previous form. My questions/suggestions to improve the MS are as follows:

1. Abstract: there should be more quantitative information included rather than only decription.

2. Material and Methods: there is no information about the statistical analysis

3. Results: Statistics are very descriptive, p value is only in one place. Some other analysis would be recommended or matching some Figures together. It would be easier to analyse data if the % of people from control were next to the first group to compare. Figures should be prepared in some other format rather than in Excel. Thay would be more professional.

4. The limitations are very short. One should be added about the smal sample of people in the MRONJ group. 

5. Conclusions: they are too elaborated.

Author Response

Dear Reviewer,

We are pleased with the opportunity to revise and resubmit our manuscript.

Considering the reviewers’ comments, all have been considered very important and were taken into profound consideration. Manuscript changes are highlighted in the revised manuscript.

Our point-by-point responses to all comments are outlined and detailed below. We hope that you find our responses satisfying.

We hope the revised manuscript will better suit the Reproduction.

We are happy to consider further revisions and we thank you for your continued interest in our research.

1. The article presents the study related to diysglicemia and medication-related osteonecrosis of the jaw. Alsthough, in the pandemic of diabetes and its negative consequances on the health quality, also oral, the article needs some corrections and cannot be published in the previous form. My questions/suggestions to improve the MS are as follows:

  1. Abstract: there should be more quantitative information included rather than only decription.  Answer: Thank you for this point, we modified the abstract.

2. Material and Methods: there is no information about the statistical analysis

Answer: Thank you for this point. We added this information in the text.

3. Results: Statistics are very descriptive, p value is only in one place. Some other analysis would be recommended or matching some Figures together. It would be easier to analyse data if the % of people from control were next to the first group to compare. Figures should be prepared in some other format rather than in Excel. Thay would be more professional.

Answer: Thank you for this point. We added this information in the text.

4. The limitations are very short. One should be added about the smal sample of people in the MRONJ group. 

Answer: Thank you for this point. We added this information in the text.

5. Conclusions: they are too elaborated.

Answer: Thank you for this point. We modified the conclusions.

Reviewer 2 Report

The manuscript “Association Between Dysglycemia and Medication-Related Osteonecrosis of the Jaw (MRONJ)” is a case-control study whose aim is “to disseminate knowledge and thereby improve patients' quality of life and prevent MRONJ”.

The paper falls within the scope of the journal.

English language and style are fine/minor spell check is required.

In my opinion, the paper possesses some flaws, in detail:

-       In the title, it is defined the term “dysglycemia”, however, it seems that the study is only on patients affected by “hyperglycemia”. If it is so, the title should be revised.

-       Introduction, some references are out of date and should be changed or associated with more recent and appropriate references (e.g., the definition of MRONJ in the first lines). Overall, this section is not focused on the topic, it describes what are the BPs and denosumab, but there is little information on MRONJ. There are several sentences that should be revised (e.g., there are several articles on the relationship between MRONJ and diabetes). Additionally, the aims seem a little bit exaggerated (i.e., “to disseminate knowledge and thereby 61 improve patients' quality of life and prevent MRONJ”).

-       M&M, how was MRONJ diagnosed and staged? Why are the stages of MRONJ not described?

Regarding the measurement of fasting glucose levels, what time of day were they performed? Did the authors ask patients to avoid something before the text? (e.g., eating or drinking)

Several risk factors have been related to MRONJ (e.g., dental extraction), however, it seems that the authors have not investigated any.

-       Results, this section, and the related tables/figures should be strongly revised. Especially, tables/figures need to be modified or removed, they do not add any relevant information to the readers (e.g., tables 1 and 2 describe the results on “prevalence of hyperglycemia and normoglycemia” in the 2 study groups, but are separated and formatted in different ways. Or table 4, its formatting needs to be reviewed, furthermore, some lines are pointless).

-       Discussion, like the rest of the manuscript, needs to be revised (e.g., why is there a question in lines 198-199?)

-       Conclusion, in my opinion, the authors did not prove what they stated (e.g., “dysglycemia significantly 230 increases the risk of developing MRONJ”). In this observational study, the authors just tested the glucose level of patients affected by MRONJ at admittance, this is not enough to prove that these two conditions are related.

Author Response

Dear Reviewer,

We are pleased with the opportunity to revise and resubmit our manuscript.

Considering the reviewers’ comments, all have been considered very important and were taken into profound consideration. Manuscript changes are highlighted in the revised manuscript.

Our point-by-point responses to all comments are outlined and detailed below. We hope that you find our responses satisfying.

We hope the revised manuscript will better suit the Reproduction.

We are happy to consider further revisions and we thank you for your continued interest in our research.

The manuscript “Association Between Dysglycemia and Medication-Related Osteonecrosis of the Jaw (MRONJ)” is a case-control study whose aim is “to disseminate knowledge and thereby improve patients' quality of life and prevent MRONJ”.

The paper falls within the scope of the journal.

English language and style are fine/minor spell check is required.

In my opinion, the paper possesses some flaws, in detail:

  •       In the title, it is defined the term “dysglycemia”, however, it seems that the study is only on patients affected by “hyperglycemia”. If it is so, the title should be revised. Answer: Thank you, we changed this part.

-       Introduction, some references are out of date and should be changed or associated with more recent and appropriate references (e.g., the definition of MRONJ in the first lines). Overall, this section is not focused on the topic, it describes what are the BPs and denosumab, but there is little information on MRONJ. There are several sentences that should be revised (e.g., there are several articles on the relationship between MRONJ and diabetes). Additionally, the aims seem a little bit exaggerated (i.e., “to disseminate knowledge and thereby 61 improve patients' quality of life and prevent MRONJ”).

  •       M&M, how was MRONJ diagnosed and staged? Why are the stages of MRONJ not described? Answer: Thank you, we changed this part and clarified in the text.

Regarding the measurement of fasting glucose levels, what time of day were they performed? Did the authors ask patients to avoid something before the text? (e.g., eating or drinking)  Answer: Thank you, we changed this part. Before blood sampling, patients were fasted for 12 hours, so the samples were taken on an empty stomach. This was facilitated by the fact that blood was taken in the morning and by qualified medical staff.

Several risk factors have been related to MRONJ (e.g., dental extraction), however, it seems that the authors have not investigated any.

Thank you for this point, We added this to the text.

  •       Results, this section, and the related tables/figures should be strongly revised. Especially, tables/figures need to be modified or removed, they do not add any relevant information to the readers (e.g., tables 1 and 2 describe the results on “prevalence of hyperglycemia and normoglycemia” in the 2 study groups, but are separated and formatted in different ways. Or table 4, its formatting needs to be reviewed, furthermore, some lines are pointless).
  • Answer: Thank you, we changed this part. 

    The data in Table 1.2 and Diagram 1.2 are presented as Table 1. Table 4 has been removed and the data presented in text: Approximately 38.3% (n = 23) of patients who were examined received anti-resorptive medical treatment due to breast carcinoma (cc), 23.3% (n = 14) anti-resorptive medical treatment due to prostate cc., 15% (n = 9) anti-resorptive medical treatment due to osteoporosis, 10% (n = 6) anti-resorptive medical treatment due to multiple myeloma, 8.3% (n = 5) anti-resorptive medical treatment due to lung carcinoma and 5% (n = 3) anti-resorptive medical treatment due to other malignant diseases (non-Hodgkin’s lymphoma, urothelial cc., and colorectal cc.). The number of patients who received bisphosphonate therapy due to breast and prostate cc. was the highest.

  •       Discussion, like the rest of the manuscript, needs to be revised (e.g., why is there a question in lines 198-199?)
  • Answer: Thank you, we corrected the text.

-       Conclusion, in my opinion, the authors did not prove what they stated (e.g., “dysglycemia significantly 230 increases the risk of developing MRONJ”). In this observational study, the authors just tested the glucose level of patients affected by MRONJ at admittance, this is not enough to prove that these two conditions are related. Answer: Thank you we agree. We changed the text in the conclusion.

Round 2

Reviewer 1 Report

Thank you for improving your manuscript, however there are still some minor elements which needs corrections:

1. Methods; Line 203-219: The description of statistical analysis is in one paragraph with the information about the Helsinki Declaration etc. Please, add some paragraphs.

2. Results; Line 241: If there is no statistical significance so there should be p>0.05. Also, please short the p value to max. 3 numbers after coma.

3. Conclusion are still too long. Maybe you can add some of these paragraph into the Discussion? Conlcusions sould only summerize the paper in one paragraph. 

Author Response

Answer to Reviewer 1.

We are pleased with the opportunity to revise and resubmit our manuscript.

Considering the reviewers’ comments, all have been considered very important and were taken into profound consideration. Manuscript changes are highlighted in the revised manuscript.

Our point-by-point responses to all comments are outlined and detailed below. We hope that you find our responses satisfying.

We hope the revised manuscript will better suit the Reproduction.

We are happy to consider further revisions and we thank you for your continued interest in our research.

-----------------------

Question: Thank you for improving your manuscript, however there are still some minor elements which needs corrections:

Methods; Line 203-219: The description of statistical analysis is in one paragraph with the information about the Helsinki Declaration etc. Please, add some paragraphs.

Answer: Thank you for this point. We corrected.

Question: Results; Line 241: If there is no statistical significance so there should be p>0.05. Also, please short the p value to max. 3 numbers after coma.

Answer: Thank you for this point. We corrected.

Question: Conclusion are still too long. Maybe you can add some of these paragraph into the Discussion? Conlcusions sould only summerize the paper in one paragraph. 

Answer: Thank you for this point. We corrected.

Reviewer 2 Report

The reviewer thanks the Authors for the point-by-point response.

However, in my opinion, the paper possesses some flaws, in detail:

- there are several typo errors (e.g. in the abstract “hypglycemia”);

- some statements are not clear, not common, incorrect, or not justified by the references (e.g. “denosumab injection (INJ)” is a very uncommon acronym; or in the conclusion the term “dysglycemia” reappears);

- usually, the aims are at the end of the introduction section, moreover, the aims of the study are still doubtful;

- conclusions are still too strong, and also too long.

Author Response

Answer to Reviewer 2.

We are pleased with the opportunity to revise and resubmit our manuscript.

Considering the reviewers’ comments, all have been considered very important and were taken into profound consideration. Manuscript changes are highlighted in the revised manuscript.

Our point-by-point responses to all comments are outlined and detailed below. We hope that you find our responses satisfying.

We hope the revised manuscript will better suit the Reproduction.

We are happy to consider further revisions and we thank you for your continued interest in our research.

-----------------------

The reviewer thanks the Authors for the point-by-point response.

However, in my opinion, the paper possesses some flaws, in detail:

Question: - there are several typo errors (e.g. in the abstract “hypglycemia”);

Answer: Thank you for this point. We corrected.

Question:- some statements are not clear, not common, incorrect, or not justified by the references (e.g. “denosumab injection (INJ)” is a very uncommon acronym; or in the conclusion the term “dysglycemia” reappears);

Answer: I appreciate this argument. We eliminated INJ. With all due respect, we would like to maintain dysglycemia in its current form, as the definition states that it is an abnormal glycemic status. (REF)

REF: Aramendi I, Burghi G, Manzanares W. Dysglycemia in the critically ill patient: current evidence and future perspectives. Rev Bras Ter Intensiva. 2017 Jul-Sep;29(3):364-372. doi: 10.5935/0103-507X.20170054. PMID: 29044305; PMCID: PMC5632980.

- usually, the aims are at the end of the introduction section, moreover, the aims of the study are still doubtful;

Answer: Thank you for this point. We modified the introduction section with the specific aim of the study.

- conclusions are still too strong, and also too long.

Thank you for making this point. We shortened the conclusion to make it clearer and more efficient.